# Effect of Mortalin on Scar Formation in Human Dermal Fibroblasts and a Rat Incisional Scar Model

**DOI:** 10.3390/ijms23147918

**Published:** 2022-07-18

**Authors:** Bok Ki Jung, Tai Suk Roh, Hyun Roh, Ju Hee Lee, Chae-Ok Yun, Won Jai Lee

**Affiliations:** 1Institute for Human Tissue Restoration, Department of Plastic & Reconstructive Surgery, Severance Hospital, Yonsei University College of Medicine, Seoul 03722, Korea; jpk1213@yuhs.ac (B.K.J.); rohts@yuhs.ac (T.S.R.); nh1005@yuhs.ac (H.R.); 2College of Medicine, Graduate School, Yonsei University, Seoul 03722, Korea; 3Department of Dermatology, Cutaneous Biology Research Institute, Severance Hospital, Yonsei University College of Medicine, Seoul 03722, Korea; juhee@yuhs.ac; 4Department of Bioengineering, College of Engineering, Hanyang University, Seoul 04763, Korea; chaeok@hanyang.ac.kr

**Keywords:** scar, keloid, mortalin, adenovirus, interleukin-1α receptor, fibrogenesis

## Abstract

Wound healing is a complicated cascading process; disequilibrium among reparative processes leads to the formation of pathologic scars. Herein, we explored the role of mortalin in scar formation and its association with the interleukin-1α receptor using in vitro and in vivo models. To investigate the effects of mortalin, we performed an MTT cell viability assay, qRT-PCR, and Western blot analyses, in addition to immunofluorescence and immunoprecipitation studies using cultured fibroblasts. A rat incisional wound model was used to evaluate the effect of a mortalin-specific shRNA (dE1-RGD/GFP/shMot) Ad vector in scar tissue. In vitro, the mortalin-treated human dermal fibroblast displayed a significant increase in proliferation of type I collagen, α-smooth muscle actin, transforming growth factor-β, phospho-Smad2/3-complex, and NF-κB levels. Immunofluorescence staining revealed markedly increased mortalin and interleukin-1α receptor protein in keloid tissue compared to those in normal tissue, suggesting that the association between mortalin and IL-1α receptor was responsible for the fibrogenic effect. In vivo, mortalin-specific shRNA-expressing Ad vectors significantly decreased the scar size and type-I-collagen, α-SMA, and phospho-Smad2/3-complex expression in rat incisional scar tissue. Thus, dE1-RGD/GEP/shMot can inhibit the TGF-β/α-SMA axis and NF-κB signal pathways in scar formation, and blocking endogenous mortalin could be a potential therapeutic target for keloids.

## 1. Introduction

Wound healing, a complicated cascading process, involves local and systemic responses that include hemostasis, inflammation, proliferation, and tissue remodeling. Normal linear scars occur through a normal wound healing response in equilibrium. However, disequilibrium among reparative processes can impair wound healing, resulting in pathologic scars with excessive connective tissue deposition (e.g., hypertrophic scars or keloids) or inadequate healing leading to chronic wounds [1,2,3,4,5,6,7]. Keloid formation involves multiple factors, including molecular signaling as well as genetic, environmental, and anatomical factors. It is related to the dysregulation of apoptosis and results from a prolonged proliferative stage and a delayed remodeling stage. Keloid formation is correlated with excessive accumulation of extracellular matrix protein, reduced apoptosis, and extracellular matrix degradation and involves numerous cytokines and growth factors [8]. Interleukin (IL)-1 and IL-6 secreted by fibroblasts may be involved in keloid pathogenesis as major proinflammatory cytokines. As a result of skin damage, the production of IL-1α by keratinocytes can stimulate IL-6 secretion from the surrounding fibroblasts [9]. Both IL-1 and IL-6 may enhance inflammation by recruiting immune cells to the area where the tumor is developing [10]. Stimulation of the malignant properties of epithelial and cancer cells by IL-1 and IL-6 is directly related to the ability of these cytokines to activate proto-oncogenic transcription factors, such as signal transducer and activator of transcription 3 (STAT3) and nuclear factor kappa-light-chain-enhancer of activated B cells (NF-κB), which exert angiogenic, immunosuppressive, and antiapoptotic effects in the tumor microenvironment.

Mortalin (Mot; mtHsp70/PBP74/Grp75) is a heat un-inducible member of the Hsp70 family of proteins and is composed of 679 amino acids (MW 73,913 Da). It plays an essential role in importing substrates into the mitochondria, responding to oxidative stress, regulating mitochondrial membrane potential, generating energy, transporting ligands into cells, protecting against apoptosis, and controlling p53 function [11,12]. In our previous study, mortalin-specific small hairpin (sh)RNAs (dE1-RGD/GFP/shMot) were generated and introduced into keloid spheroids [13]. The results showed that the proliferative and anti-apoptotic functions of mortalin are associated with the pathogenesis of keloids via p53 and the transforming growth factor (TGF)-β1/Smad pathway [13]. Furthermore, certain studies have revealed that mortalin interacts with the IL-1α receptor and is involved in receptor-mediated internalization [14]. The complex of mortalin and IL-1α receptor might be trafficked to the nucleus [14]. Then, the translocated receptor complex with appropriate intranuclear binding proteins might increase or decrease the transcriptional activity of NF-κB [14]. However, no study has explored the association between mortalin and IL-1 on the formation of pathologic scars.

As gene-delivery vehicles, adenovirus (Ad)-based vectors are widely used in experiments because of their high transduction efficiency and titer capability regardless of the cell division stage [15,16,17]. However, a replication-incompetent Ad vector system that knocked-down mortalin (Ad-shMot) is limited in animal models, owing to immunogenicity, local inflammation, enzymatic inactivation, short half-life, and transient effects. To address this, polyphthalamide (PPA) can be used as a depot delivery system for the slow-release of Ad and the prolongation of Ad activity in the experimental environment [15,16,17,18].

Patients with keloids suffer from various symptoms such as pain and pruritus, but a clear treatment method has not yet been identified. Interleukins are important factors in the mechanism of keloid formation. In particular, the role of IL-1 involved in the mechanism of mortalin is a growth factor that mainly contributes to the formation of keloids. Therefore, in this study, we explored the role of cytosolic and extracellular mortalin in keloid pathogenesis using cultured human dermal fibroblasts (HDFs) and keloid fibroblasts (KFs). We evaluated the fibrosis in the HDFs and KFs with extracellular treatment of exogenous mortalin. The expression of endogenous mortalin in keloid tissues and normal tissues was compared and confirmed by correlating with the molecular mechanism related to IL-1.

Keloid scarring is unique to humans. Furthermore, because the physiology of skin and the immune system of animals are vastly different from those of humans, animal models may not be reliable for preclinical studies on keloids [19,20]. Although the mechanisms of keloid and scar formation may be different, the aim of this study was to investigate the influence of mortalin on the inflammation process as a common wound healing process, and consequently, its effect on scar formation, overall. Therefore, in this study, we explored the anti-fibrotic effects of a mortalin-specific shRNA (dE1-RGD/GFP/shMot) on scar formation in a rat incisional scar model.

## 2. Results

### 2.1. In Vitro Study

#### 2.1.1. Mortalin Acts as Profibrotic Molecules in HDFs

The enhanced and sustained release of mortalin-induced profibrotic effects in HDFs. HDFs were cultured in the presence of 100 ng mortalin and proliferation was evaluated via a 3-(4,5-dimethylthiazol-2-yl)-2,5-diphenyltetrazolium bromide (MTT) assay. Each experiment was repeated three times. Cell proliferation was significantly increased in the mortalin-treated group when compared with the level in the untreated HDFs group (* *p* < 0.05; Table 1 and Figure 1a), indicating that mortalin enhanced cell viability. Moreover, these results were presumed to be similar to the effects of TGF-β1, acting as a known profibrotic cytokine.

We investigated whether mortalin can induce the synthesis and deposition of collagen and compared the effect of mortalin with that of TGF-β1 on collagen synthesis in normal HDFs using quantitative real-time reverse transcriptase-polymerase chain reaction (qRT-PCR). The mRNA expression of type I collagen in HDFs was increased to similar levels following treatment with either mortalin (50 and 100 ng) or TGF-β1 (10 ng) (* *p* < 0.05; Figure 1b).

Western blotting confirmed these results. HDFs were treated with 100 ng/mL mortalin for 72 h. In the mortalin-treated group, the protein levels of collagen types I and III were significantly increased (* *p* < 0.05; Figure 1c,e). Also, we evaluated the expression level of α-smooth muscle actin (α-SMA) for whether mortalin could differentiate HDFs into myofibroblasts. As shown in Figure 1d,f, a significant increase in the expression level of α-SMA was found in the presence of 100 ng of mortalin.

#### 2.1.2. Mortalin Increased Intracellular Signaling via TGF-β1 and NF-κB

We investigated the association of mortalin-induced collagen synthesis with activation of TGF-β1/pSmad signaling. Western blot was performed to evaluate the expression of profibrogenic TGF-β1 and pSmad 2/3 complex after treatment with mortalin (100 ng). In the results, the expression of TGF-β1 and pSmad2/3 complex increased following treatment with mortalin (100 ng) (Figure 2a). The protein expression level of TGF-β and pSmad2/3 complex significantly increased following treatment with mortalin (* *p* < 0.05; Figure 2c,d). In addition, the expression of NF-κB, as intracellular signaling molecules involved in collagen synthesis and myofibroblast differentiation, was significantly elevated following mortalin treatment (* *p* < 0.05; Figure 2b,e). Therefore, these results demonstrated that the profibrogenic effect of mortalin was mediated via the activation of the TGF-β1/pSmad signaling pathway.

#### 2.1.3. Mortalin Activates the IL-1α Receptor, Interacts with the IL-1α Receptor in the Cytosol, and Internalizes the IL-1α Receptor in Keloid Tissues

Immunofluorescence staining was performed to evaluate the expression of mortalin and IL-1α receptor protein in keloid tissues and normal tissues. Compared with the expression in normal tissues, the expression of mortalin and IL-1α receptor protein immunoreactivity was increased markedly in keloid tissues (Figure 3a,b).

As shown in Figure 3c,d, mortalin and IL-1α receptor protein were observed on the HDFs in the perinuclear area; however, they were overexpressed on the KFs, and localized to both the nucleus and entire cytoplasm. The cytoplasmic accumulation of IL-1α receptor was particularly notable in keloid tissues.

KFs were immunoprecipitated with anti-mortalin-antibody and analyzed by Western blotting. As shown in Figure 4a, mortalin and IL-1α receptor interacted with each other and mortalin was bound to the IL-1α receptor in KFs.

Based on the semi-quantitative MetaMorph^®^ image analysis, both the expression of mortalin and IL-1α receptor were significantly increased as a result of immunoprecipitation of mortalin in KFs. It can be hypothesized that mortalin binds to the IL-1a receptor. In the case of additional TGF-β treatment, the effect was found to be much greater, and it can be inferred that the response of mortalin and IL-1a receptor is increased by TGF-β on keloid formation (Figure 4b,c).

Mortalin binds to the IL-Iα receptor on membranes of fibroblasts and is involved in the internalization of mortalin and IL-1α receptor complex. The mortalin and IL-1α receptor complex might be trafficked and the IL-1α receptor translocated to the nucleus, where it subsequently induces IL-1 signaling. Thereafter, the fibrosis reaction may be amplified leading to keloid scar formation.

### 2.2. In Vivo Study

#### 2.2.1. Mortalin-Specific shRNA-Expressing Ad Vectors Decreased Scar Size in Rat Incisional Scar Model

The effect of a mortalin-specific shRNA-expressing Ad on the expression of extracellular matrix (ECM) components of scars was confirmed through histological examination. As a result of hematoxylin and eosin (H&E) and Masson’s trichrome (MT) staining, re-epithelialization was completed, and scar and granulation tissue were observed in all groups on day 14 of the postoperative period (Figure 5a,b). At 14 days postoperatively, active inflammation with abundant inflammatory cells and immature collagen fibers was observed in the C-group, but the MV-group exhibited a lower level of collagen fiber and the infiltration of fewer inflammatory cells in the scar area of H&E-stained tissues. MT staining of specimens revealed that collagen type I deposition was decreased in the MV-group compared with that in the other groups. In addition, dense and coarse collagen bundle structures were replaced by immature collagen bundles in the MV-group.

To evaluate the scar area and degree of granulated tissue formation, only the boundary of the scar area between the epidermis and the panniculus carnosus was measured following MT staining. The area of the scar with granulation tissue was quantitatively measured in two different MT-stained specimens within the same wound. Quantitative analysis of the scar area revealed that the mean ± standard error of the mean (SEM) scar sizes in C-, P-, CV-, and MV-groups were 53,081.3 ± 6946.6, 55,161.3 ± 8190.1, 51,483.0 ± 1532.7, and 40,708.2 ± 6564.7 µm^2^, respectively, on day 14 of the postoperative period. These results indicated that shMot-expression by Ad (dE1-RGD/GFP/shMot) reduced the size of scars compared with those of the other groups, respectively (* *p* < 0.05; Figure 5c).

#### 2.2.2. Mortalin-Specific shRNA-Expressing Ad Decreases Collagen Type I Expression in Rat Incisional Scar Tissue

Type I collagen synthesis was analyzed among the four groups using Western blot (Figure 6a). On day 14 postoperation, collagen type I protein expression was significantly decreased in scar tissue in the MV-group compared with those in the C-, P-, and CV-groups (* *p* < 0.05, Figure 6b).

The results suggested that the expression of collagen type I protein, as a major ECM component, decreased following the injection of mortalin-specific shRNA-expressing Ad in rat incisional scar tissue.

#### 2.2.3. Mortalin-Specific shRNA-Expressing Ad Decreases the Expression of α-SMA and pSmad2/3 Complex in Rat Incisional Scar Tissue

α-SMA is a marker of the effects of TGF-β on the wound healing pathway. Western blot (Figure 7) analyses were performed to examine the expression of α-SMA and pSmad2/3 complex. The expression of α-SMA and the pSmad2/3 complex was significantly decreased in rat scar tissues from the MV-group compared with that in rat scar tissues from the C-, P-, and CV-groups (* *p* < 0.05).

## 3. Discussion

Keloids are hyperplastic pathological scars that extend beyond the boundaries of the original wound and continue to grow slowly, like a skin tumor [21,22]. Traditionally, hypertrophic scars and keloids have been diagnosed as two distinct diseases; however, the differences between the two types of pathologic scars have not yet been clarified [23,24,25]. Through the sustained upregulation of highly sensitive proinflammatory genes, traumatic and inflammatory stimuli trigger the formation of the keloid scars [24,25,26]. Some researchers have argued that keloids are simply more aggressive forms of hypertrophic scars and that the two can be considered continuing stages of the same fibroproliferative skin disease, with varying degrees of inflammation [27]. Inflammation is essential in the early stages of normal wound healing [28,29,30]. Nonetheless, an inappropriately excessive or delayed inflammatory phase that can begin as an acute reaction may lead to pathological scars [27]. Although several studies have shown that inflammation in pathologic scar formation involves a wide range of complex mechanisms, such as angiogenesis [31], neurogenic inflammation [32], and other unknown reactions, this still remains to be elucidated [27].

The unclear pathogenesis of keloids is partially responsible for the paucity of effective treatments and the unpredictable prognosis for patients. Many studies have shown that the pathophysiology of keloids is due to prolonged proliferation and delayed remodeling [33,34] and is caused by an excessive accumulation of ECM [3]. Importantly, it is revealed by the dysregulation of the signaling pathway by TGF-β [8,35,36,37], as a proinflammatory cytokine. Evidence to date suggests that inflammation triggers the subsequent immune response cascade and that ILs are associated with keloid formation [9,10,38]. ILs are major inflammatory factors and potentially regulate the recruitment, proliferation, differentiation, and apoptosis of fibroblasts, and the production of ECM. IL-1 is important in the initial stage of keloid formation. IL-1 receptor antagonist (IL-1RA) is a member of the IL-1 gene family, that binds to the IL-1 receptor and specifically blocks the activity of IL-1 [39]. In a previous report, the administration of IL-1RA to a New Zealand rabbit model was shown to effectively reduce skin fibrosis [40]. In addition, after hyperbaric oxygen therapy, the expression of IL-1RA was increased in keloids [41], which lowered the level of inflammation and presented a better therapeutic effect.

Mortalin (mot-2/mtHsp70/PBP74/GRP75) is an essential protein in the Hsp70 family of chaperones [11]. The functions of mortalin include mitochondrial import, intracellular trafficking via association with the IL-1 or FGF-1 receptor, and inactivation of p53, and it has a role in the regulation of cell proliferation [11]. Although studies have investigated the p53-related mechanism of mortalin for the treatment of cancer [42,43,44,45], research on the underlying mechanism of mortalin on pathologic scars (keloid or hypertrophic scar) [13] or on other functions, such as chaperonization and intracellular trafficking via association with the IL-1 receptor, is limited.

In the present study, a keloid scar was applied in the in vitro study, whereas an incisional scar was used in the in vivo study, and both studies were carried out separately and simultaneously. Although distinct mechanisms of scar formation could be at work in the two scar types examined, the aim of the present study was to investigate the influence of mortalin in the inflammation process, and in turn, in overall scar formation. In our study, the treatment of HDFs with exogenous mortalin resulted in a fibrogenic effect, which was similar to that observed following treatment with TGF-β. The proliferative viability of HDFs and the accumulation of proteins related to fibrogenesis (collagen I and α-SMA) increased following treatment with exogenous mortalin. In our study, the expressions of TGF-β, signaling molecules such as NF-κB, and pSmad2/3 complex were significantly increased in HDFs following treatment with exogenous mortalin. These results suggested that the overexpression of exogenous mortalin in normal HDFs acted as a fibrogenic cytokine, upregulating signaling pathways underlying keloid scar formation.

In addition, our results showed that mortalin and IL-1α receptor expression was increased in keloid tissues compared with that in adjacent normal tissues. Under normal conditions, mortalin is distributed in all cell types, especially in the perinuclear area. IL-1α receptor is expressed in various cell types, including fibroblasts and cancer cells [9,10]. However, mortalin was overexpressed in the cytosol of keloid tissue; the distribution of IL-1α receptor was similar to that of mortalin. The results suggest that overexpressed mortalin in the cytosol interacted with the IL-1α receptor and triggered the fibrogenic cascade in keloid tissue. Considering the results of the immunoprecipitation study, mortalin and IL-1α receptor were prepared as a complex. Mortalin was bound to IL-1α receptor, IL-1α receptor was localized in the cytoplasm, and the IL-1α receptor trafficked from the cytosol to the nucleus. Translocation of the IL-1α receptor induced transcriptional activity mediated by NF-κB, which contributed to the inhibition of apoptosis, accelerating the division of mutant cells and keloid scar formation.

The present study, which was based on a rat incisional scar model, focused on the influence of mortalin in the inflammatory phase and the fibrosis process during conventional and keloid scar formations. Based on the results, we hypothesized that reducing mortalin overexpression might exert an anti-fibrotic effect. We generated mortalin-specific shRNAs (dE1-RGD/GEP/shMot), which were injected into the scar tissues of a rat incisional scar model to confirm the anti-fibrotic effect. In our study, dE1-RGD/GEP/shMot decreased scar size and the deposition of granulated tissue and collagen, which were observed microscopically. Following the injection of dE1-RGD/GEP/shMot into the scar, Western blot revealed that the expression of collagen type I, the major ECM component, was significantly decreased. dE1-RGD/GEP/shMot revealed a significant decrease in the expression of α-SMA and NF-κB in the scar tissue. TGF-β levels in KFs were increased concomitantly with increased α-SMA expression and induced α-SMA expression, suggesting that α-SMA expression may be a marker of TGF-β activity [46,47,48]. Activation of the NF-κB pathway plays a role in keloid formation by preventing KFs from undergoing apoptosis [49,50,51]. Notably, in dermal cells, such as dermal keratinocytes and fibroblasts, NF-κB is essential to promote inflammation and wound healing in response to trauma; these findings support the role of NF-κB in the formation of keloids [51,52]. Thus, these results suggested that dE1-RGD/GEP/shMot could inhibit the TGF-β/SMA axis and NF-κB signal pathways for the prevention of scars. Furthermore, overexpression of mortalin has a critical role in keloid pathogenesis. Therefore, inhibition of mortalin expression could be a promising therapeutic target for the treatment of keloids or hypertrophic scars.

## 4. Materials and Methods

To investigate the effects of exogenous and endogenous mortalin, we performed an MTT cell viability assay, qRT-PCR, and Western blot analyses, in addition to immunofluorescence and immunoprecipitation studies using cultured HDFs and KFs.

A rat incisional wound model was used to evaluate the effect of a mortalin-specific shRNA (dE1-RGD/GFP/shMot) Ad vector in scar tissue. Four groups of animals were used as follows: phosphate-buffered saline (PBS) injection group (C-group), PPA group (P-group), control virus group (CV-group), and shMot virus group (MV-group). On day 14 after surgery, rats were sacrificed, and a tissue biopsy was performed. Subsequently, histological (H&E and MT staining) examinations, an enzyme-linked immunosorbent assay (ELISA) for mortalin expression, and Western blot analysis were performed.

### 4.1. In Vitro Studies

#### 4.1.1. Keloid Tissue, HDF, and Normal Abdominal Tissue

Human keloid tissue samples were obtained from patients and healthy donors according to protocols approved by the Yonsei University College of Medicine with the Institutional Review Board (IRB). Written informed consent was obtained from the patients prior to sample collection. Keloid tissues from patients with active-stage keloid (*n* = 5) and normal skin tissues from the healthy donors (abdomen, thigh, and back) (*n* = 5) were obtained by excision for fibroblast culture and histological and immunofluorescence analyses. All experiments involving human tissues were carried out according to the guidelines of the Declaration of Helsinki. Primary HDFs and KFs were obtained from the American Type Culture Collection (ATCC; Manassas, VA, USA). Cells used in this study were cultured in Dulbecco modified Eagle medium (DMEM; GIBCO, Grand Island, NY, USA) supplemented with 10% heat-inactivated fetal bovine serum (FBS), penicillin (100 U/mL), and streptomycin (100 μg/mL).

#### 4.1.2. Cell Viability Assay

An MTT assay was performed to validate cell viability for proliferation and metabolic activity. HDFs (5 × 10^4^ cells/cm^2^) were exposed to 10 ng of TGF-β and 100 ng of mortalin for 48 h. Next, the cells were incubated at 37 °C in fresh culture medium in an incubator with 5% CO_2_, and the culture medium was removed. Then, 200 μL of MTT solution (0.5 mg/mL in PBS; Boehringer, Mannheim, Germany) was added to each well, and the cells were incubated at 37 °C for 3 h. After removing the MTT solution to dissolve the precipitates, 200 μL of dimethyl sulfoxide was added. The substrate medium was removed and 200 μL of dimethyl sulfoxide solution (Sigma-Aldrich, St. Louis, MO, USA) was added to each well, and then the OD was read at 570 nm using an ELISA reader (Bio-Rad, Hercules, CA, USA).

#### 4.1.3. qRT-PCR

HDFs (2 × 10^5^ cells) were treated with TGF-β (10 ng) or mortalin (50 or 100 ng). After 48 h, total RNA was extracted with the RNeasy Mini kit (Qiagen, Hilden, Germany), and complementary DNA was prepared from 0.5 g of total RNA using a first-strand cDNA synthesis kit (AccuPower, RT PreMix, Bioneer, Daejeon, Korea). Applied Biosystems TaqMan primer/probe kits were used to analyze mRNA expression levels with an ABI Prism 7500 HT Sequence Detection System (Applied Biosystems, Foster City, CA, USA).

#### 4.1.4. Western Blot Analysis

Samples were lysed in 50 mM Tris-HCl (pH 7.6), 1% Nonidet P-40(NP-40), 150 mM NaCl, and 0.1 mM zinc acetate in the presence of protease inhibitors. The quantitative protein level in the solution was estimated through the Lowry protein assay (Bio-Rad, Hercules, CA, USA). A 30 g protein sample was separated using 10% sodium dodecyl sulfate-polyacrylamide gel electrophoresis (SDS-PAGE). The protein in the solution was electrotransferred to a polyvinylidene fluoride membrane, which was subsequently incubated with primary antibodies against mortalin, IL-1α receptor, collagen type I, collagen type III, α-SMA, NF-κB, TGF-β, pSmad2/3 complex, and β-actin. Membranes were then incubated with a secondary antibody conjugated to horseradish peroxidase (HRP) anti-rabbit or anti-mouse (Southern Bio Technology Associates, Inc., Birmingham, AL, USA). Protein expression patterns were visualized using an enhanced chemiluminescence detection kit (sc-2004; Santa Cruz Biotechnology, Santa Cruz, CA, USA) and analyzed using ImageJ software (National Institutes of Health, Bethesda, MD, USA). We used primary mouse anti-mortalin monoclonal antibody (mAb [C1-3)), rat anti- IL-1α receptor antibody, mouse anti-collagen type-I mAb (Abcam, Cambridge, UK), mouse anti-collagen type-III mAb (Sigma-Aldrich, St. Louis, MO, USA), rabbit anti-TGF-β1 mAb (Abcam), rabbit anti-β-actin antibody (Sigma-Aldrich, St. Louis, MO, USA), rabbit anti-pSmad 2/3 mAb (Cell Signaling Technology, Beverly, MA, USA), and rabbit anti-actin mAb (Sigma-Aldrich, St. Louis, MO, USA).

#### 4.1.5. Immunofluorescence Assay

Cultured cells (KFs, HDFs, HDFs treated with TGF-β) and specimens (normal & keloid tissues) were washed twice with PBS and fixed in 4% paraformaldehyde for 10 min at room temperature and permeabilized by incubating for 15 min in 0.01% Tween^®^ 20 in PBS. The samples were blocked with 5% bovine serum albumin and incubated with mouse anti-mortalin monoclonal (C1-3) and rat anti-IL-1α receptor primary antibody overnight at 4 °C. After one day, cells were washed with PBS, and incubated with Alexa Fluor 488-conjugated goat anti-rabbit IgG (Invitrogen, Life Technologies, Grand Island, NY, USA) and Alexa Fluor 594-conjugated goat anti-mouse secondary antibody (Invitrogen) for 2 h at room temperature. The cells were plated on slides using Vectashield^®^ mounting medium (Vector Laboratories, Burlingame, CA, USA) with 4′,6-diamidino-2-phenylindole (DAPI) (Vector Laboratories Inc., Burlingame, CA, USA) and observed with a confocal microscope (LSM700, Olympus Corp., Center Valley, PA, USA).

#### 4.1.6. Immunoprecipitation Assay

KFs were washed, pelleted, and resuspended in lysis buffer supplemented with protease inhibitors (20 mm Tris-HCl, pH 7.5, 150 mm NaCl, 10% glycerol, and 1% Triton X-100). Cell lysate was precleared, and supernatants were incubated overnight with anti-IgG or anti-mortalin antibody on a rotating platform at 4 °C, and then incubated with protein A-Sepharose Fast Flow beads. Beads were collected, washed, and resuspended in an equal volume of 5× SDS loading buffer. Immunoprecipitated proteins were separated using 12% SDS-PAGE. Western blotting was performed using the appropriate antibody in the manner previously described. Mortalin and Il-1a receptor levels were semi-quantitatively analyzed using the MetaMorph^®^ image analysis software (Universal Image Corp., Buckinghamshire, UK). Results are expressed as the mean optical density of six different digital images per sample.

### 4.2. In Vivo Rat Incisional Scar Model

#### 4.2.1. Animal Model

Incisional wounds were studied in 25 male Sprague–Dawley rats. The animal experiments were approved by the Institutional Animal Care and Use Committee of Yonsei University (#2018-0219, approved by 16 July 2020). General anesthesia was induced via an intraperitoneal injection of a mixture of zolazepam tiletamine (30 mg/kg, Zoletil^®^; Virbac, Carros, France) and xylazine (10 mg/kg, Rompun^®^; Bayer, Leverkusen, Germany). An 8 × 1 cm^2^ rectangle section of skin, subcutaneous fat, and muscle were excised with full thickness and only the skin layer was closed to maximize tension by leaving the muscle unsutured (Figure 8).

#### 4.2.2. Generating shMot-Expressing Adenoviral Vectors

Replication-incompetent Ad expressing shMot (dE1-RGD/GFP/shMot; Ad-shMot) and control Ad (dE1-RGD/GFP/scramble; Ad-scramble) were used in this study [13]. To improve the transduction to KFs compared to the wild-type Ad, both replication-incompetent Ads possessed RGD, a tripeptide composed of glycine, l-arginine, and l-aspartic acid that recognizes subtypes of integrins in fibers by providing an alternative viral entry pathway into KFs [53,54,55]. To generate Ad expressing GFP and shMot or scramble at the E1 and E3 regions, respectively, pdE1-RGD/GFP [56] was linearized by SpeI digestion and co-transformed into *Escherichia coli* BJ5183 with the XmnI-digested pSP72-E3/CMV-shMot or -scramble E3 shuttle vector [57] for homologous recombination. This generated either a pdE1-RGD/GFP/shMot or scramble Ad vector. Ad was propagated, purified, and titrated as described previously [58,59] (Figure 9).

#### 4.2.3. Injection of Ad into the Rat Incision Model

After surgery, the experimental animals were randomly assigned to one of four treatment groups: C-group, control group—PBS (8000 μL) injection (*n* = 5); P-group, PPA (PPA 100 μL and PBS 7900 μL) injection (*n* = 5); CV-group, control virus group—control virus with PPA complex injection (*n* = 5); and MV-group, shMot virus group—shMot virus with PPA complex injection (*n* = 10). Rats were injected with 1 mL of PBS, the same amount of PPA only, control virus with PPA complex injection, or shMot virus [5 × 10^9^ plaque/mL] with PPA complex, respectively. Injections were administered using a 1 mL syringe with a 27-gauge needle directly into the intradermal layers of the surgical sites, 0, 1, and 3 days after surgery.

#### 4.2.4. Histologic Analysis

Twenty-five Sprague–Dawley rats were euthanized on day 14, and tissue biopsies were obtained. The specimens (thickness of 10 mm) were collected from the mid-section of the scar area where the skin tension was the greatest. All tissues were fixed in 10% neutral-buffered formalin, embedded in a paraffin block, and subjected to H&E and MT staining. In order to estimate the scar area and the degree of tissue granulation, the tissues stained with H&E and MT were examined under an optical microscope at 40× magnification.

To estimate the scar area, only the boundary of the scar area between the epidermis and the panniculus carnosus was measured. The scar areas with the granulated tissue were measured from two different MT-stained specimens in the same wound. Data for each measurement are shown as the mean ± SEM. The scar area was estimated using Image J software version 1.49 (National Institutes of Health, Bethesda, MD, USA). For each wound, the mean scar area was then converted from pixel numbers to square micrometers, calculated using the ratio of pixel numbers to the scale bar.

#### 4.2.5. Western Blot Analysis

Samples were lysed in 50 mM Tris-HCl (pH 7.6), 0.1 mM zinc acetate, 150 mM NaCl, 1% Nonidet P-40, and protease inhibitors. Total protein concentration was quantified using the Lowry method (Bio-Rad), and 3 g of sample was separated using 10% SDS-PAGE. The proteins on the gel were electrotransferred to a polyvinylidene fluoride membrane, and incubated with primary antibodies against mortalin, collagen type I, collagen type III, α-SMA, pSmad2/3 complex, and β-actin. Samples were then incubated with a secondary antibody conjugated to HRP anti-rabbit or anti-mouse (Santa Cruz Biotechnology). The expression patterns were determined using an ECL detection kit (sc-2004; Santa Cruz Biotechnology). Protein expression was analyzed using the ImageJ software (National Institutes of Health). We used primary anti-collagen type-I mAb (Abcam), mouse anti-collagen type-III mAb (Sigma-Aldrich, St. Louis, MO, USA), rabbit anti-pSmad 2/3 mAb (Cell Signaling Technology, Beverly, MA, USA), and Cru anti-β-actin mAb (Sigma-Aldrich, St. Louis, MO, USA).

### 4.3. Statistical Analysis

The results are presented as the mean and standard deviation. Data were analyzed using repeated one-way analysis of variance (ANOVA). Two sets of independent data were compared using a paired *t*-test; a difference was considered significant at *p* < 0.05 (SPSS for Windows 26.0).

## 5. Conclusions

Keloids are pathologic scars that result from prolonged proliferation phase and delayed remodeling phase and are associated with profibrogenic cytokines, such as TGF-β, ILs, and other factors. Mortalin is a member of the heat shock protein 70 family of chaperones, which exists in the nucleus, intracellular, and extracellular spaces of all cell types, and is involved in intracellular trafficking association with the IL-1α receptor during the regulation of cell proliferation. We confirmed that an overexpression of exogenous or endogenous mortalin induced fibrogenesis and the internalization of IL-1α receptor and exerted a fibrogenic effect on HDFs. Moreover, knockdown of mortalin induced an anti-fibrotic effect in a rat incisional scar model. In conclusion, blocking endogenous mortalin may represent a potential therapeutic target for keloid scars.

## Figures and Tables

**Figure 1 ijms-23-07918-f001:**
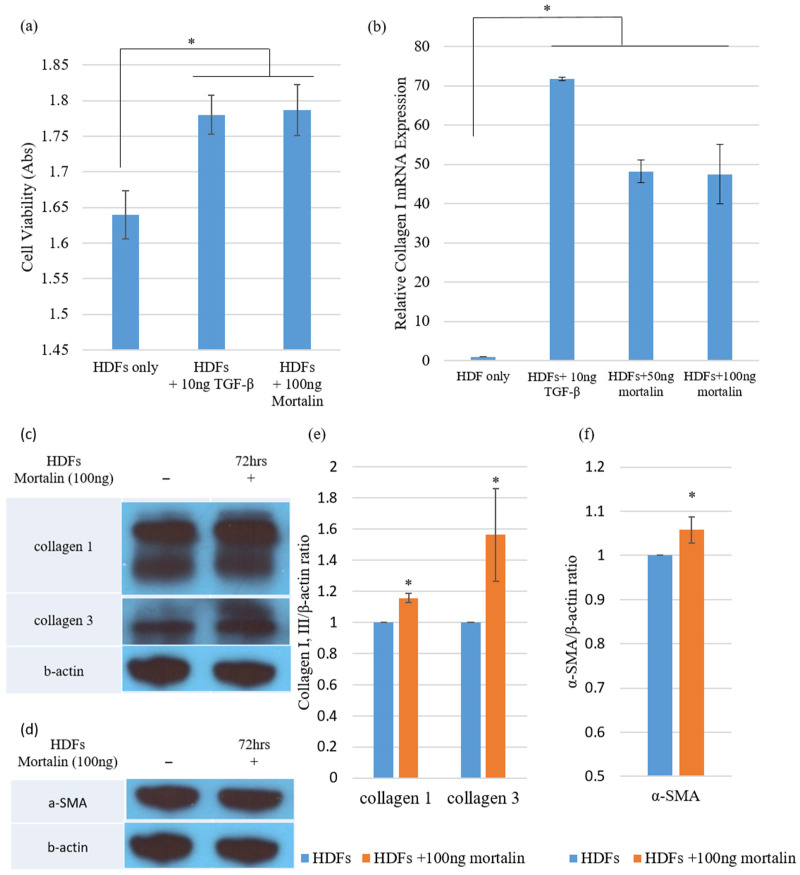
Mortalin functions as a profibrotic molecule. (**a**) A significant increase in proliferation activity was found in mortalin-treated HDFs via an MTT assay (100 ng; * *p* < 0.05); (**b**) The mRNA expression of type I collagen was significantly increased in HDFs to comparable levels following treatment with mortalin (50 or 100 ng) or TGF-β1 (10 ng) (* *p* < 0.05); (**c**,**d**) The levels of collagen type I, collagen type III, and α-SMA were determined by Western blotting using HDF lysates treated with mortalin (100 ng) for 72 h; (**e**) Protein expression levels of collagen types I and III were significantly increased after 72 h (* *p* < 0.05); (**f**) Stimulation of HDFs with mortalin significantly increased the protein expression of α-SMA (* *p* < 0.05). Data are expressed as mean ± SD (*n* = 5). * *p* < 0.05 indicates statistically significant differences as compared with the control group.

**Figure 2 ijms-23-07918-f002:**
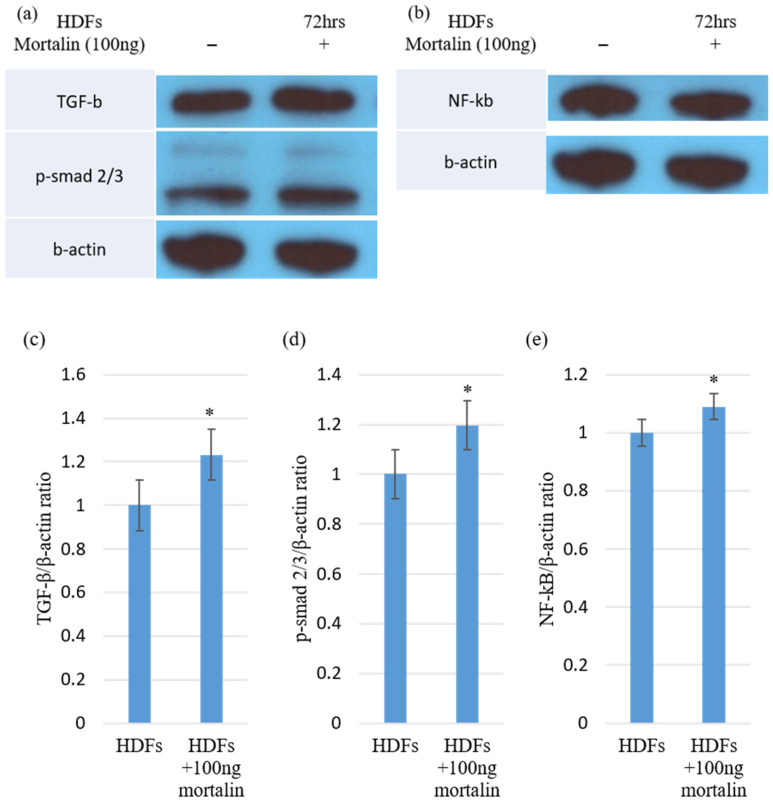
Mortalin induced TGF-β, Smad, and NF-κB in HDFs. (**a**,**b**) The levels of TGF-β, pSmad2/3 complex, and NF-κB proteins were determined using Western blotting in HDFs lysates treated with mortalin (100 ng); (**c**) A significant increase in TGF-β1 expression was observed in the mortalin (100 ng)-treated HDFs (* *p* < 0.05); (**d**) pSmad2/3 complex protein levels were significantly increased in mortalin (100 ng)-treated HDFs (* *p* < 0.05); (**e**) NF-κB protein levels were significantly increased in mortalin (100 ng)-treated HDFs when compared with non-treated HDFs (* *p* < 0.05). Data are expressed as mean ± SD (*n* = 5). * *p* < 0.05 indicates statistically significant differences as compared with the control group.

**Figure 3 ijms-23-07918-f003:**
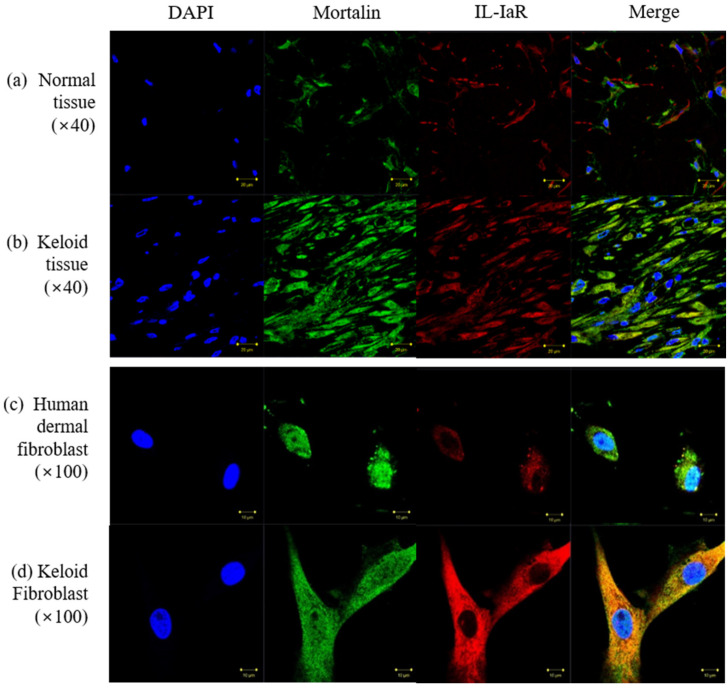
Evaluation of mortalin expression using immunofluorescence (mortalin, green; nucleus, blue; IL-1α receptor, red) (**a**,**b**) Mortalin and IL-1α receptor were overexpressed in keloid tissues compared to their expression in the extra-lesional normal tissues (40×); (**c**,**d**) Mortalin and IL-1α receptor were expressed in the perinuclear area on the HDFs; however, overexpression of mortalin and IL-1α receptor in the cytosol was observed on the KFs, and cytoplasmic accumulation of IL-1α receptor was observed in the KFs (100×). HDFs and KFs were double labeled with mortalin and IL-1aR; Scale bars: 10 µm for tissue and 20 µm for fibroblast.

**Figure 4 ijms-23-07918-f004:**
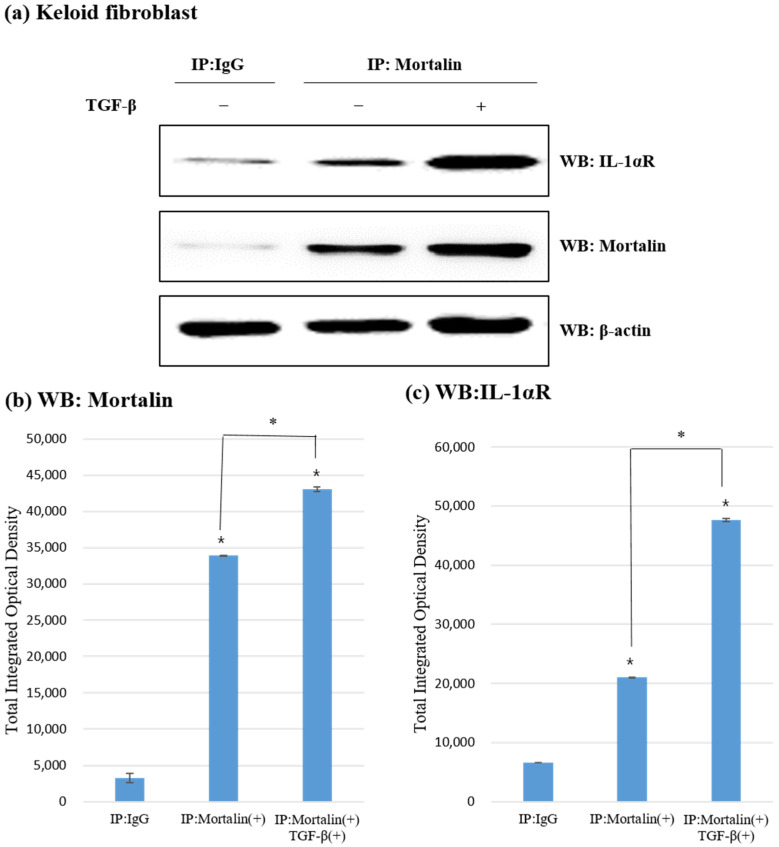
Immunoprecipitation and Western blotting analysis of mortalin and IL-1α receptor in keloid fibroblasts. (**a**) In KFs, mortalin and IL-1α receptor interact with each other and mortalin was bound to the IL-1α receptor. The mortalin and IL-1α receptor levels were higher in KFs than in HDFs and were further upregulated following the activation of KFs by TGF-β1 (10 ng/mL); (**b**,**c**) On the semi-quantitative MetaMorph^®^ image analysis, expression of mortalin and the IL-1a receptor was significantly increased as a result of immunoprecipitation of mortalin in KFs (* *p* < 0.05) and was significantly increased upon additional TGF-β treatment (* *p* < 0.05).

**Figure 5 ijms-23-07918-f005:**
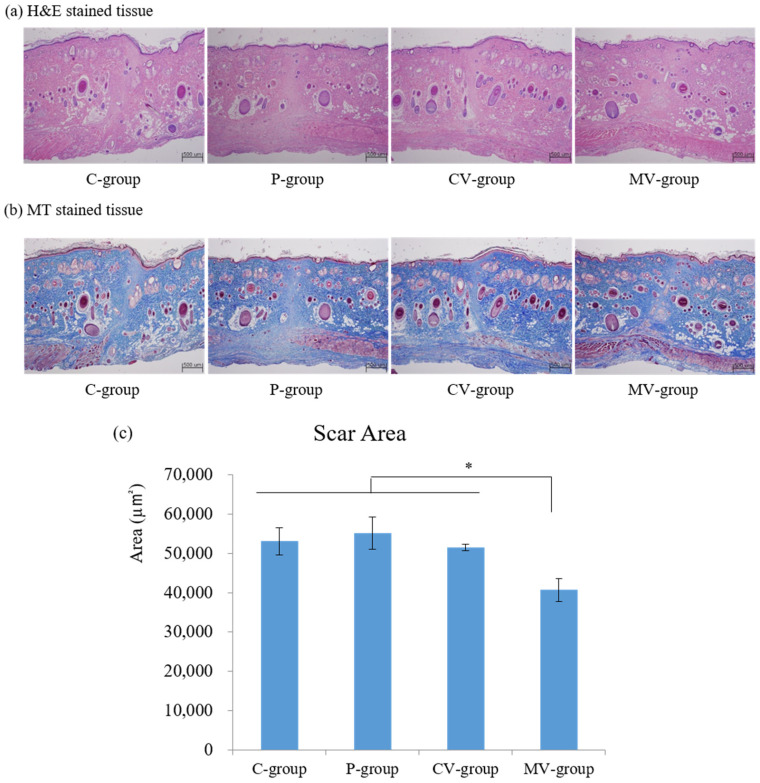
H&E and MT-stained tissues from the C-, P-, CV-, and MV-groups in rat incisional scar model on day 14 (40×). (**a**) H&E and (**b**) MT-stained tissues from the C-, P-, CV-, and MV-groups on day 14 (magnification, 40×). The C-group continued to present a wide area of granulated tissue with inflammation. The MV-group presented lower levels of immature collagen deposition within scar areas; (**c**) The scar area was significantly narrower in the MV-group than in the C-, P-, and CV-groups on day 14. (* *p* < 0.05). C-group, control group-PBS injection; P-group, PPA injection; CV-group, control virus with PPA complex injection; MV-group, shMot virus with PPA complex injection. Data are expressed as mean ± SD (*n* = 5). * *p* < 0.05 indicates statistically significant differences as compared with the control group.

**Figure 6 ijms-23-07918-f006:**
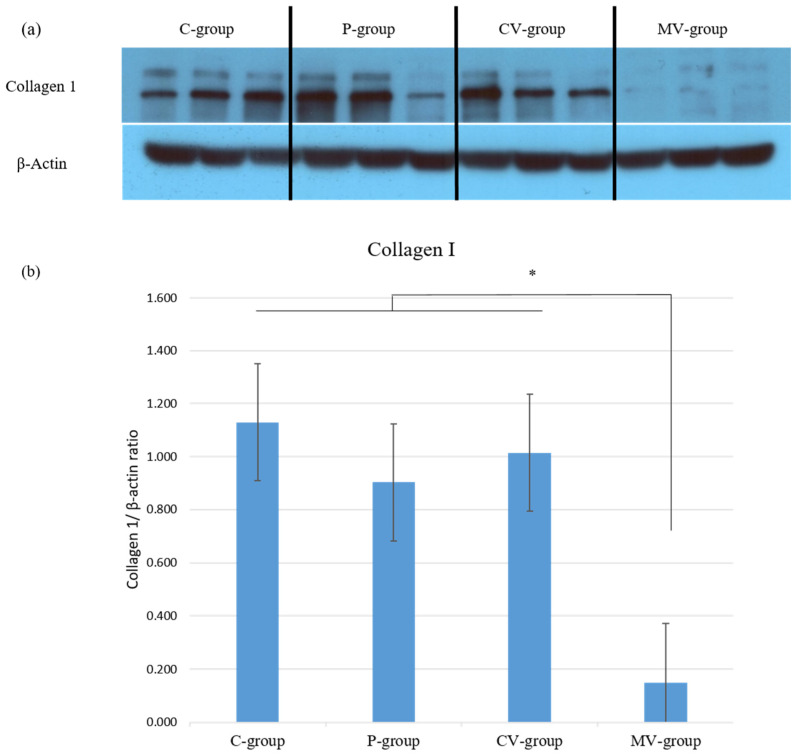
Mortalin-specific shRNA-expressing Ad decreased the expression of collagen type I in rat incisional scar model. (**a**) Collagen type I expression was evaluated among the four groups using Western blotting; (**b**) The expression of collagen type I was significantly decreased in the MV-group versus the C-, P- and CV-groups (* *p* < 0.05). Data are expressed as mean ± SD (*n* = 5). * *p* < 0.05 indicates statistically significant differences as compared with the control group.

**Figure 7 ijms-23-07918-f007:**
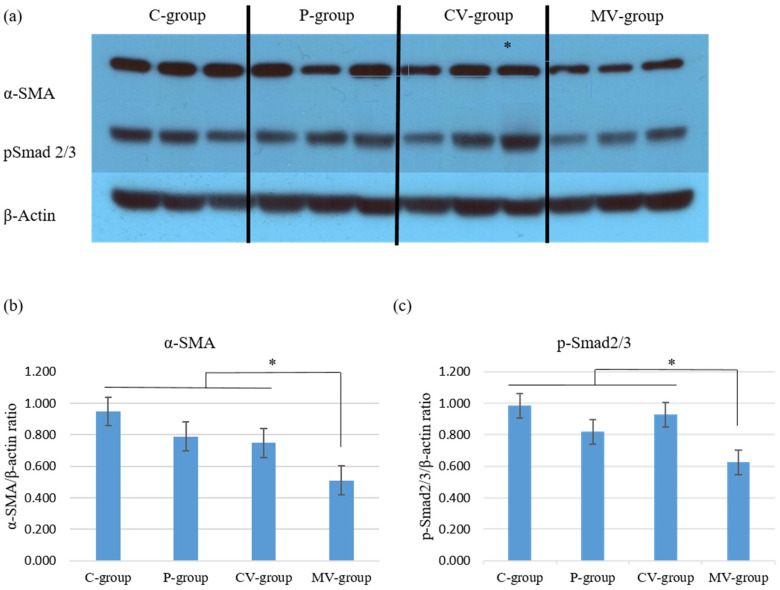
Mortalin-specific shRNA-expressing Ad decreased the expression of α-SMA and the p-Smad2/3 complex in rat incisional scar model. (**a**) Expression of α-SMA and the p-Smad2/3 complex among the four groups was determined using Western blotting; (**b**,**c**) The expression of α-SMA and the p-Smad2/3 complex was significantly decreased in the MV-group compared with that in the C-, P- and CV-groups (* *p* < 0.05). Data are expressed as mean ± SD (*n* = 5). * *p* < 0.05 indicates statistically significant differences as compared with the control group.

**Figure 8 ijms-23-07918-f008:**
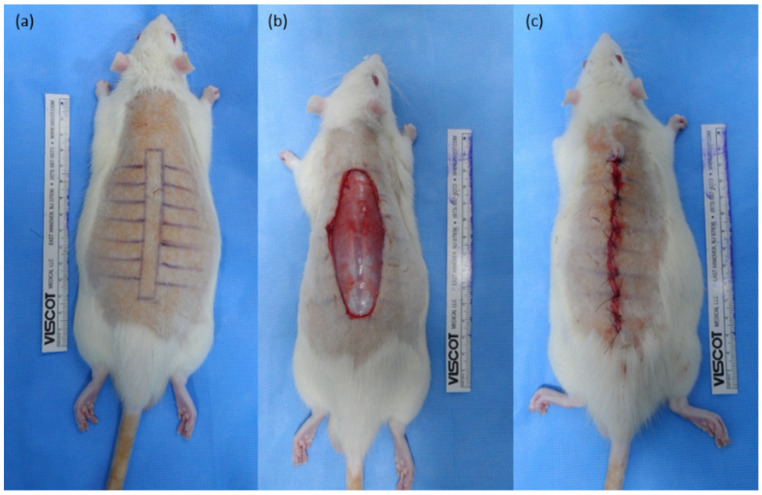
Rat incisional scar model. (**a**) An 8 × 1 cm^2^ rectangular cut was made on the dorsal skin of a male Sprague–Dawley rat; (**b**) The entire skin and muscle were excised; (**c**) The skin layer was sutured, leaving the muscle layer unsutured to maximize the tension in the skin sutured area.

**Figure 9 ijms-23-07918-f009:**
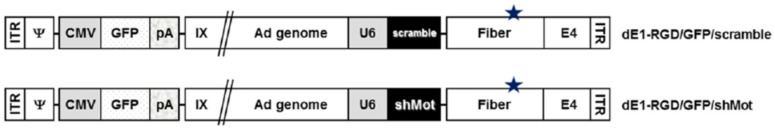
Schematics of the shMot-expressing adenoviral vectors. The RGD-incorporated adenoviruses were generated by inserting the RGD motif between HI loop of the fiber knob (star). (ITR = inverted terminal repeat; Ψ = packaging signal; pA = polyA sequence; IX = protein IX; shMot = mortalin-specific small hairpin (sh)RNAs).

**Table 1 ijms-23-07918-t001:** Cell proliferation activity of mortalin using MTT assay.

Group	MTT Assay	
	Mean ± SD	(% of Control)
Control (HDFs only)	1.64 ± 0.083	100
HDFs + 10 ng TGF-β1	1.78 ± 0.067	108.54
HDFs + 100 ng mortalin	1.79 ± 0.088	108.96

Normal HDFs were incubated with 10 ng TGF-β1 or 100 ng mortalin for 48 h. The value represents the mean ± SD for triplicate experiments.

## Data Availability

The data that support the findings of this study are available on request from the corresponding author. The data are not publicly available due to privacy or ethical restrictions.

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
