# Peer review of "Effect of Mortalin on Scar Formation in Human Dermal Fibroblasts and a Rat Incisional Scar Model"

_ijms, 2022, doi:10.3390/ijms23147918_

Round 1

Reviewer 1 Report

All points were adressed.

However, in my opinion some figures need some layout revisions . The standard excel graph layout does not look fine  (e.g. bigger numbers , in black not grey), bigger y-axis designations, better axis devision, to much zeros in the numbers). In figure 5 c there are also an unnessary decimal.) In figure 5 there are  2 photographs , to small and without labelling.

Author Response

Thank you for reviewing this manuscript.

As you advised, I have made corrections to the figures.

I changed to bigger numbers, black letters, and unnecessary number after decimal point was deleted.

Also, I deleted the figures without labelling in figure 5.

Reviewer 2 Report

This study topic is interesting. Overall, this report has good quality and the authors have provided some results to support the significance of this study. Reasonable revisions are needed before acceptance.

Comments and suggestions:

1, the reason for choosing/designing this study need to be explained more

2, more results beyond western blot are suggested for this report

3, more background and refs about wound healing are suggested to be cited/discussed, such as: Acta Biomaterialia, 147, (15),2022, 147-157;  ACS Applied Materials & Interfaces ,2022,14 (6), 7680-7689;  Advanced Functional Materials, 2022,2202410

4, the illustration figure is needed

5, the language need to be improved and double checked

Author Response

  • Thank you for reviewing this manuscript.
  1. As you advised, I have added the reason for this study in introduction section, in the previous version.
  2. I agreed your opinion. In order to obtain good contents about the purpose of the experiment, various methods of experiment were conducted; MTT assay, qRT-PCR, western blot, immunofluorescence assay and immunoprecipitation study using human dermal fibroblast and keloid fibroblast and tissue staining, western blot using tissue from rat incisional scar model.
  3. As you mentioned, I added the references in the introduction section.
  4. I agreed with your opinion that the illustration will make easier to understand.

      5. English grammar has been reviewed by experts in the previous version.

This manuscript is a resubmission of an earlier submission. The following is a list of the peer review reports and author responses from that submission.

Round 1

Reviewer 1 Report

This study topic is interesting and important. Overall, this report has good quality and the authors have provided some results to support the significance of this study. Reasonable revisions are needed before acceptance.

Comments and suggestions:

1, the reason for choosing/designing this study need to be explained more

2, an illustration figure about this study is needed

3, more characterization tests for the materials are needed for this report 

4, more background and refs about wound repair are suggested to be cited/discussed, such as: Journal of nanobiotechnology,2021, 19 (1), 1-12; Biomaterials Science,2021, 9 (5), 1530-1546; Chemical Engineering Journal,2020, 392, 123775

5 the language need to be double checked

Author Response

Thank you for your review of our manuscript titled ‘The effect of mortalin on scar formation in human dermal fibroblasts and a rat incisional scar model’.

It is a great honor to be given the chance to be commented on by the renowned reviewers of your highly respected and prestigious journal.

We are happy to respond to the detailed review of the respected reviewer’s.

The detailed answers of the review was added as a attached file;

Reviewer 2 Report

Dear author,

Your study is interesting and show some interesting hints regarding keloids. However, you missed to show important things to support you thesis and many questions remain unanswered for example:

  1. You have tested mortalin on HDFs but not on KFs. (which cells did you used for your experiments:isolated cells or those from ATCC)
  2. Can you see differences between HDF and KFs?
  3. Why you did not show more results with TGF as a kind of positive control?
  4. Why did you not use the in vitro system to knock down mortalin expression? Which should be easier than in an animal model?
  5. Do you think 10-20 % more protein expression in western blots could statistically significant with only n=5? Also is an increase of 10-20% biological significant? In comparison to TGF?
  6. Why did you not try a quantitative analysis of the immunofluorescence or FACS analysis to support your conclusion? Also a HDF immunoprecipitation and WB would be helpful
  7. It is not really clear how you obtained your results (s.below). So, estimation for the reader is diffult. Here, did the animals show macroscopic differences in scar developments?
  8. Is it now worth a try to overexpress mortalin?
  9. Is it possible that the obtained effects in animals may be mediated by the reduction of skin cells via reduction of mortalin? à in vitro experiments mortalin knock down?

Minor points:

Abstract: Methods decription (human?) cell culture? Cells ued Keloid tissue? Patient Nr.?Mortalin?

To much details for adenovirus

Introduction: humaans.

Results:

 Figure 1: How many experiments? Mean+/-sd?

Fig.1b ->collaten?  The significance * are badly placed. Mean+/-sd? No sd marks.

Fig.1 C,e,f  TGF as positive control?

Fig 1C,d protein bands seem overexposed, the cut is not so good, more space!

fig 1 e, f normalized on control? Statistic tests on raw data?

Legend: No conclusion in the legend “…mortalin increased the protein expression of alpha sma…”

  • The increase of alpha-SMA seems very low (<10%)! And this in a semi quantitative western blot! Comparison to TGF? (Normally TGF at least double the signal) Not enough culture time?

Fig 2: How many experiments? Mean+/-sd?

In the text you decribed: mRNA expression of TGF-β and pSmad2/3 com-123 plex significantly increased following treatment with mortalin (*p < 0.05; Figures 2c and 124 2d).

In the legend of fig.2  it seems that you show protein levels?

fig 2 c,d f normalized on control? Statistic tests on raw data?

Fig. 2 a,b b-actin seems the same as in fig.1. à same blot? Band pictures are badly cut.

  • Comparison to TGF
  • Low impact of mortalin on TGF pathway

Fig 3 Normal tissue pictures with significant less cells than keloid tissue!

How many experiments?

  • Please digital analysis of the fluorescence signals, difficult by eye because more cells more signal!
  • In addition FACS for better quantification!

Fig. 4 Where are the HDF samples? Where are the quantifications?

Fig.5 Legend doe not described the experiments.

 For example skin samples of rat, mice or human??  

Fig 5c How many experiments? Mean +/-sd? Y-axis? Decimal?  Please explain how the scar area was defined! -> show it on the MT slides.Methods section is unclear.

Pictures of scars macroscopic?

Fig.6 and 7 It is not clear how the results were obtained (From tissue, where it was locqted or isolated cells) and how many samples were used finally. (n? mean+/-sd, should be explained also in the legend)

Material/Methods:

Primary cells isolated from patients (normal/keloid)  and also form American Type culture collection?

Did you test the mortalin-specific shRNA in cell culture?

4.2.5. To use 30 g of a sample is not usual! How you collected here the material for western blot? Did you isolate cells? How many tissue?

Author Response

(The authors gave the same response as above.)
